

# NO₃ reactivity during a summer period in a temperate forest below and above the canopy

Patrick Dewald[1], Tobias Seubert[1], Simone T. Andersen[1], Gunther N. T. E. Türk[1], Jan Schuladen[1], Max R. McGillen[2], Cyrielle Denjean[3], Jean-Claude Etienne[3], Olivier Garrouste[3], Marina Jamar[4], Sergio Harb[5], Manuela Cirtog[5], Vincent Michoud[6], Mathieu Cazaunau[5], Antonin Bergé[5], Christopher Cantrell[5], Sebastien Dusanter[4], Bénédicte Picquet-Varrault[5], Alexandre Kukui[7], Chaoyang Xue[1,7], Abdelwahid Mellouki[2,8], Jos Lelieveld[1], and John N. Crowley[1]

[1]Atmospheric Chemistry Department, Max Planck Institute for Chemistry, 55128 Mainz, Germany
[2]Institut de Combustion, Aérothermique, Réactivité Environnement (ICARE), CNRS, 1C Avenue de la Recherche Scientifique, CEDEX 2, 45071 Orléans, France
[3]CNRM, Université de Toulouse, Meteo-France, CNRS, Toulouse, France
[4]IMT Nord Europe, Institut Mines-Télécom, Université de Lille, Center for Energy and Environment, 59000 Lille, France
[5]Université Paris Est Créteil and Université de Paris Cité, CNRS, LISA, F-94010 Créteil, France
[6]Université Paris Cité and Université Paris Est Créteil, CNRS, LISA, F-75013 Paris, France
[7]Laboratoire de Physique et Chimie de l'Environnement et de l'Espace (LPC2E), CNRS, Orléans, France
[8]University Mohammed VI Polytechnic (UM6P), Lot 660, Hay Moulay Rachid Ben Guerir, 43150, Morocco

*Correspondence to*: John N. Crowley (john.crowley@mpic.de)

**Abstract.** We present direct measurements of BVOC-induced nitrate radical (NO₃) reactivity ($k^{VOC}$) through the diel cycle in the suburban, temperate forest of Rambouillet near Paris (France). The data were obtained in a six-week summer period in 2022 as part of the ACROSS campaign (Atmospheric ChemistRy Of the Suburban foreSt). $k^{VOC}$ was measured in a small (700 m²) clearing mainly at a height of 5.5 m above ground level, but also at 40 m (for 5 days/nights). At nighttime, mean values of $k_{night}^{VOC}$(5.5 m) = (0.24 ± 0.27) s⁻¹ and $k_{night}^{VOC}$(40 m) = (0.016 ± 0.007) s⁻¹ indicate a significant vertical gradient and low NO₃ reactivity above the canopy, whereas $k_{night}^{VOC}$(5.5 m) showed peak values of up to 2 s⁻¹ close to the ground. The strong vertical gradient in NO₃ reactivity could be confirmed by measurements between 0 and 24 m on one particular night characterised by a strong temperature inversion, and is a result of the decoupling of air masses aloft from the ground- and canopy-level sources of BVOCs (and NO). No strong vertical gradient was observed in the mean daytime NO₃ reactivity with $k_{day}^{VOC}$(5.5 m) = (0.12 ± 0.04) s⁻¹ for the entire campaign and $k_{day}^{VOC}$(40 m) = (0.07 ± 0.02) s⁻¹ during the 5-day period.

Within the clearing, the fractional contribution of VOCs to the total NO₃ loss rate ($L^{NO_3}$, determined by photolysis, reaction with NO and VOCs) was 80-90 % during the night and ~50 % during the day. In terms of chemical losses of α-pinene below canopy height in the clearing, we find that at nighttime OH and O₃ dominate with NO₃ contributing "only" 17 %, which decreases further to 8.5 % during the day. Based on OH, O₃ and NO₃ concentrations, the chemical lifetime of BVOCs at noon is about one hour and is likely to be longer than timescales of transport out of the canopy (typically in the order of minutes), thus significantly reducing the importance of daytime, in-canopy processing. Clearly, in forested regions where sufficient NOₓ




is available, the role of NO$_3$ and OH as initiators of BVOC oxidation are not strictly limited to the night and to the day,
respectively, as often implied in e.g. atmospheric chemistry text-books.

## 1 Introduction

Forests emit great quantities ($\sim$ 1000 Tg yr$^{-1}$) of a variety of biogenic volatile organic compounds (BVOCs) such as isoprene
and monoterpenes into the atmosphere (Guenther et al., 2012; Hakola et al., 2012; Vermeuel et al., 2023). The transport of
combustion-related emissions from urban and industrialised regions results in the presence of NO$_X$ (the sum of nitric oxide,
NO and nitrogen dioxide, NO$_2$) in forested regions, as does microbial activity in soils (Ludwig et al., 2001; Barger et al., 2005;
Pilegaard, 2013). An important step in photochemical ozone (O$_3$) generation is the oxidation of VOCs, which may be initiated
by hydroxyl (OH) and nitrate radicals (NO$_3$) or O$_3$ itself (Geyer et al., 2001; Lelieveld et al., 2008; Peräkylä et al., 2014). The
interaction of largely anthropogenic NO$_X$ with BVOCs is thus a key component of tropospheric ozone production in many
regions (Pusede et al., 2015). Here we focus on NO$_3$, which is formed from the reaction between NO$_2$ (e.g. from R1) and O$_3$
(R2) and is in a thermal equilibrium with NO$_2$ and dinitrogen pentoxide (N$_2$O$_5$, R3 and R4) (Wayne et al., 1991).

$$NO + O_3 \rightarrow NO_2 + O_2 \tag{R1}$$
$$NO_2 + O_3 \rightarrow NO_3 + O_2 \tag{R2}$$
$$NO_3 + NO_2 + M \rightarrow N_2O_5 + M \tag{R3}$$
$$N_2O_5 + M \rightarrow NO_3 + NO_2 + M \tag{R4}$$

In the troposphere, NO$_3$ reacts efficiently with NO to re-form NO$_2$ (R5), reacts with unsaturated VOCs (R6) and is photolysed
rapidly with a lifetime that is often only a few seconds (R7a and R7b) (Finlayson-Pitts and Pitts, 2000). NO is reduced in
concentration at night owing to its reaction with O$_3$ and since both NO and sunlight drastically reduce the lifetime of NO$_3$, the
latter is often thought of as a "nighttime only" oxidant (Wayne et al., 1991; Platt and Heintz, 1994; Martinez et al., 2000;
Brown and Stutz, 2012). Note that, in some environments, direct heterogeneous losses of NO$_3$ can also be important as can
indirect losses via N$_2$O$_5$ uptake (Saathoff et al., 2001; Bertram and Thornton, 2009; Phillips et al., 2016).

$$NO_3 + NO \rightarrow 2\ NO_2 \tag{R5}$$
$$NO_3 + VOC\ (+ O_2) \rightarrow\rightarrow products\ (e.g.\ RONO_2, HNO_3) \tag{R6}$$
$$NO_3 + h\nu \rightarrow NO + O_2 \tag{R7a}$$
$$NO_3 + h\nu \rightarrow NO_2 + O \tag{R7b}$$

While both the reaction with NO (R5) and photolysis (R7) regenerate NO$_2$ and thus recycle NO$_X$, reactions between NO$_3$ and
VOCs result in a variety of gas-phase products including organic nitrates (RONO$_2$) and nitric acid (HNO$_3$) (Hallquist et al.,
1999; Ayres et al., 2015; Ng et al., 2017) which may be lost by deposition and/or transferred to the condensed phase, forming
e.g. secondary organic aerosols (SOA) (Bates et al., 2022; Day et al., 2022; DeVault et al., 2022). The interaction of NO$_3$ with
BVOCs can represent an efficient process for the removal of NO$_X$ from the gas phase and a mechanism for SOA generation



(Fry et al., 2014; Romer Present et al., 2020), making the fractional contribution of R6 to the overall loss rate of $NO_3$ of particular interest.

As unsaturated BVOCs such as isoprene and monoterpenes are often present at parts per billion by volume (ppbv) levels in the forest (Kesselmeier and Staudt, 1999; Hakola et al., 2009) the local $NO_3$ lifetimes are typically short not only during the day, but also at night (McLaren et al., 2004; Liebmann et al., 2018a; Liebmann et al., 2018b). $NO_3$ mixing ratios are often

below 1 part per trillion by volume (pptv), making its detection in highly reactive air masses very challenging (Liebmann et al., 2018a). Measuring the $NO_3$ reactivity (together with $NO_2$ and $O_3$ to calculate the $NO_3$ production rate) provides a means to assess the atmospheric fate of the nitrate radical even when its mixing ratio is too low (< 0.5 pptv) to be detected (Dewald et al., 2022).

There are also meteorological effects that induce differences in the fate of $NO_3$ during the day and night. While daytime

insolation at ground level can result in efficient (turbulent) mixing of the boundary layer, the radiative cooling of the ground at lower temperatures at night means that the nocturnal boundary layer can be highly stratified (Stull, 1988). This results in strong gradients in the mixing ratios of e.g. BVOCs (Fish et al., 1999) and the below-canopy reactivity of $NO_3$ can be very different to that above (Mogensen et al., 2015; Liebmann et al., 2018a). To date, $NO_3$ vertical profiles are available for high altitudes (i.e. above the boundary layer) in non-forested environments (Smith et al., 1993; von Friedeburg et al., 2002; Stutz

et al., 2004; Brown et al., 2007a; Yan et al., 2021), yet highly-resolved vertical profiles of $NO_3$ (or its reactivity) for low altitudes at nighttime are sparse (Brown et al., 2007b; Liebmann et al., 2018a).

In this study, we present and analyse nighttime and daytime $NO_3$ reactivity measurements above and below the canopy in a temperate forest ca. 50 km from Paris (France) during the summer of 2022 as part of the ACROSS campaign (Atmospheric ChemistRy Of the Suburban foreSt) (Cantrell and Michoud, 2022).

## 2 Experimental

### 2.1 Site description and meteorology

The ACROSS campaign took place from mid-June until the end of July 2022 at a clearing (ca. 700 $m^2$) in the suburban forest of Rambouillet in France (N48.687, E1.704). Rambouillet forest covers an area of about 150 $km^2$ and is located about 50 km to the south-west of Paris. The surrounding trees are mainly oaks (~68 %) and pine (up to 25 %) (Marchant et al., 2017), with

an average height of ~25 m. Daytime maximum temperatures during ACROSS were between ~20–40 °C (Fig.1, panel b), while nighttime temperatures decreased to 10–25 °C, often with a significant (positive) gradient in height that started to develop in the late afternoon when tree-induced shadowing of the ground led to radiative cooling (Andersen et al, 2024).

48h-back trajectories show that air originated from the Atlantic Ocean between June 25 and July 2, whereas air passed predominantly over industrialized regions including Paris, the UK, Benelux states and the Ruhr area from July 2 to July 18

(Andersen et al., 2024). Local wind directions and speeds are shown as a wind-rose in the Supplement (Fig. S1).



The clearing housed several instrumented containers and also a tower, which enabled measurements at 40 m to be made, either via instruments located at the top of the tower or via sampling from a gas-manifold attached to a high-flow inlet at the top of the tower.

## 2.2 NO₃ reactivity

The flow-tube cavity-ring-down spectrometer (FT-CRDS, Liebmann et al. (2017)) used to quantify NO₃ reactivity ($k^{\mathrm{VOC}}$) was installed in the MPIC container and sampled mainly from the centre of a high volume-flow (10000 standard L min⁻¹, SLPM) stainless steel tube (⌀ = 15 cm) the top of which was at 5.5 m above ground level. $k^{\mathrm{VOC}}$ is defined as the VOC-induced pseudo first-order NO₃ loss rate (in units of s⁻¹) and is equal to Σ$k_i$[VOC]$_i$, with [VOC]$_i$ and $k_i$ being the VOC concentrations and the corresponding rate coefficient for R6, respectively. The FT-CRDS was connected to the high-flow inlet with a 1.5 m long

piece of ¼ inch outer diameter (OD) PFA tubing (overall residence time of 0.4 s) equipped with a Teflon membrane filter (2 µm pore, 47 mm diameter, Pall Corp.) to prevent particles entering the cavity. From July 18, the NO₃ reactivity setup sampled air alternately from the high-flow inlet and from the manifold taking air from 40 m. The instrument was attached to the manifold with ca. 20 m ¼ inch PFA tubing (ca. 5 s residence time).

NO₃ was generated by mixing 3–5 standard cm³ min⁻¹ (sccm) of NO (1 part per million by volume (ppmv) in N₂, Air Liquide)

with O₃ in a thermostated (30 °C), Teflon-coated (FEPD-121, Chemours) reactor (ca. 5 min residence time) at a pressure of 1100 Torr. O₃ was generated by the 185 nm photolysis of O₂ in a flow of 400 sccm dry synthetic air, which was provided from a commercial zero-air generator (CAP-180, Fuhr GmbH). In order to convert the resultant N₂O₅ quantitatively to NO₃ (R4), the flow was heated to 140 °C in ca. 15 cm of ¼ inch OD PFA tubing. The flow containing NO₃ was then mixed with 2.8 SLPM of either synthetic air (to define zero reactivity) or ambient air within a ¼ inch PFA T-piece and directed to the FEP-

coated, darkened and thermostated (20°C) flow-tube where NO₃ had 11 s to react. NO₃ surviving the reactor was quantified on-line by cavity-ring-down spectroscopy at 662 nm. Zeroing ("baseline measurement") was achieved by adding an excess of NO (3 sccm of 100 ppmv in N₂, Air Liquide) to titrate NO₃ (R5). In synthetic air, the NO₃ mixing ratios were typically in the range of 30–50 pptv. As the presence of NO₃ and N₂O₅ in ambient air would bias the measurement, the air was sampled through a 2 L glass flask (ca. 40 s residence time) heated to 35 °C to ensure that ambient N₂O₅ is converted to NO₃. All radicals

including NO₃, OH, RO₂ and HO₂ are lost on the glass walls and thus prevented from reaching the flowtube. Accurate quantification of NO₃ reactivity requires that the synthetic air is humidified to ambient level (monitored with a commercial sensor, IST, HYT393), which was achieved with a permeation tube (MH-070-24F-4, PermaPure LLC) filled with de-ionized water (LiChrosolv, Merck GmbH). Dynamic dilution of ambient air with synthetic air extended the dynamic range of the instrument to NO₃ reactivities of up to ca. 2.1 s⁻¹. Since reactions R1 to R5 and wall losses (0.001 s⁻¹) take place in addition to

the reaction of interest (R6), a numerical simulation procedure was used to separate contributions of NO$_X$ in the flowtube and thus extract the NO₃ reactivity towards VOCs as detailed in Liebmann et al. (2017). When sampling through the glass flask, ambient NO mixing ratios used for the simulation were corrected for the impact of R1 with correction factors between 1 and 40 %.





The uncertainty in $k^{VOC}$ is determined by the stability of the $NO_3$ source, the cavity stability (i.e. noise level and baseline

stability) and by the numerical simulation corrections. The uncertainty induced by the simulation is dependent on the ratio between $NO_2$ and $k^{VOC}$ (Liebmann et al., 2017). During the campaign source and cavity stability, numerical simulation and uncertainties in the NO correction (impact of R5) contributed on average ca. 11, 9 and 26 % to the average overall uncertainty of ca. 30 %. The limit of detection (LOD) is derived from the variability (two standard deviations, $2\sigma$) in consecutive baselines and $NO_3$ source measurements and was on average 0.006 $s^{-1}$ during this campaign. In this manuscript we index the $NO_3$

reactivity according to the following scheme: $k_{night}^{VOC}(5.5\ m)$, $k_{day}^{VOC}(5.5\ m)$, $k_{night}^{VOC}(40\ m)$ and $k_{day}^{VOC}(40\ m)$ which are the measured reactivities towards VOCs during the day/night at the two different heights. Note that reactivity measurements at 40 m was limited to the last 5 days of the campaign. We also refer to $L^{NO_3}$ which is the total $NO_3$ loss term including reaction with NO and photolysis as well as reaction with VOCs.

### 2.3 NO, NO₂ and O₃

$NO_2$ mixing ratios at 5.5 m were measured by sampling via 1.5 m ¼ inch (OD) PFA tubing and a membrane filter (2 μm pore, 47 mm diameter, Pall Corp.) through the second (405 nm) cavity of the $NO_3$ reactivity instrument (Liebmann et al., 2018b). The instrument's LOD was 87 pptv ($2\sigma$, 4 s) and the measurement was associated with a total uncertainty of 7 %. On top of the tower (40 m), $NO_2$ was measured via a cavity attenuated phase shift (CAPS) setup (precision of 6 %, LOD of 40 pptv) which was zeroed on an hourly basis.

NO was measured using a commercial instrument based on chemiluminescence detection (Ecophysics, CLD 780 TR, LOD of 10 pptv for 1 min averaging time) which was installed in a container ca. 17 m distance from the MPIC container and sampled at a height of 3.2 m above the ground. The NO mixing ratios were corrected for a change in sensitivity during the campaign (Andersen et al., 2024). Measurement of NO at 40 m height was carried out using another CLD (Teledyne, T200UP) that sampled from the tower-manifold. This measurement was corrected for losses from R1, with corrections ranging from 1–28

%. The instrument's LOD was 30 pptv and its total uncertainty was 3.2 %.

A commercial ozone monitor (2B Technologies, model 205) based on UV absorption was installed in the MPIC container and measured ozone mixing ratios at 5.5 m height with an LOD of 2 ppbv and an associated uncertainty of 5 %. $O_3$ from top of the tower was quantified with means of a second ozone monitor with an LOD of 2.5 ppbv (HORIBA, APOA370).

### 2.4 Photolysis rates and temperatures

Actinic flux was measured on top of the tower (41 m) as well as above the roof of the MPIC container (5 m), in both cases using spectral radiometers (Metcon GmbH). Actinic fluxes were converted to photolysis rates of $NO_3$ using evaluated absorption cross sections (Meusel et al., 2016). Commercial temperature sensors (Atexis PT1000 and Thermoset PT100) monitored ambient air temperature simultaneously at 5 m, 13 m, 21 m and 41 m.





## 3 Results and discussion

An overview of the measurements relevant for analysis of $NO_3$ reactivity is given in Fig. 1 where data obtained at 5.5 m (orange) and 40 m above ground level (blue) are plotted.

The VOC-induced $NO_3$ reactivity at a height of 5.5 m ($k^{VOC}$(5.5 m), panel f, orange dots) was generally high with values between ~ 0.1 and 0.5 s$^{-1}$ and also highly variable with nighttime peak values of up to 2 s$^{-1}$. In contrast, $k^{VOC}_{night}$(40 m) was often close to or below the LOD ($\leq$ 0.006 s$^{-1}$) when sampling from top of the tower between July 18 – July 23 (panel f, blue dots).

NO mixing ratios (panel e) at 3.2 m a.g.l. (orange dots) were on average between 0.1 and 0.3 ppbv, but occasionally peaked at 1–4 ppbv mostly in the morning during the continental phase. As detailed in Andersen et al. (2024), nighttime NO mixing ratios were close to or below the LOD when sampling air masses which, according to two-day back trajectories, were largely of continental origin (July 2 – July 18). In contrast, up to several ppbv of NO were observed at night during a period dominated by air with its origin over the Atlantic (June 26 – July 2) when $O_3$ levels (at 5.5 m) were below the LOD (panel c). $NO_2$ mixing

ratios (panel d) were similar at both heights and, in the absence of anthropogenic influence, mostly between 0.5 and 2 ppbv.

The large diel variation in $O_3$ mixing ratios at 5.5 m (40–80 ppbv during the day and as low as zero at nighttime, panel c) results from its net daytime photochemical production through reactions involving OH, $NO_X$ and VOCs (Crutzen and Lelieveld, 2001) and its nocturnal losses via reactions (e.g. with NO and BVOC) and deposition. Nocturnal $O_3$ mixing ratios at 40 m (20–40 ppbv) are higher than at 5.5 m (0–20 ppbv) and its diel cycle at 40 m is weaker than at 5.5 m. This results from

a combination of removal processes of $O_3$ at lower levels (reaction with NO released from soil, reaction with unsaturated, biogenic VOCs released from vegetation and deposition to soil and foliar surfaces) and weak vertical mixing at nighttime (Andersen et al., 2024).

## 3.1 $k^{VOC}$: Variability and controlling factors close to the ground

At a height of 5.5 m, $k^{VOC}$ shows large variability across the diel cycle and also from night-to-night (Fig. 1, panel f), which

are driven by variability in emissions of reactive BVOCs and in the height of the nocturnal boundary layer (NBL). In Fig. 2, $k^{VOC}$(5.5 m) is plotted together with the difference in temperature ($\Delta$T) measured at 5 and 41 m where $\Delta$T = T(5 m) – T(41 m) and shows that the largest values of $k^{VOC}_{night}$(5.5 m) occur on nights when a temperature inversion ($\Delta$T < 0 K) evolves. The impact of temperature inversion on $NO_3$ reactivity at 5.5 m is illustrated by comparing a night with no temperature inversion (July 24 – July 25, period A) to a night with moderate temperature inversion (30 June – 1 July, period B). In the absence of a

temperature inversion we see a roughly constant value < 0.1 s$^{-1}$ for $k^{VOC}$(5.5 m) through the diel cycle (period A in the inset to Fig. 2). In contrast, when a temperature inversion developed (period B in the inset to Fig. 2), $k^{VOC}$(5.5 m) was relatively low (< 0.1 s$^{-1}$) until ca. 19:30 UTC. Over the next 2 hours, it gradually increased to reach peak values as large as 0.4 s$^{-1}$, which were associated with larger variability. Over the same period $O_3$ levels decreased from ~ 30–40 ppbv to < 10 ppbv, which has been rationalised in terms of suppression of entrainment of above-canopy air (with higher levels of $O_3$) into air masses close



to the ground during temperature inversions (Andersen et al., 2024). The temperature inversion and associated reduction in vertical mixing impedes upward transport of both NO (emitted from the soil) and BVOCs (emitted from vegetation) so that both may accumulate within and below the canopy after sunset. The association of high values of $k^{VOC}$(5.5 m) with low values of $O_3$ during nighttime temperature inversions has previously been reported for the boreal forest (Liebmann et al., 2018a). In addition, elevated $NO_3$ reactivity at night is also aided by the fact that the nocturnal mixing ratios of $O_3$ and OH are diminished

due to deposition and/or lack of photochemistry, so that the lifetime and mixing ratios of monoterpenes increase (see section 3.6).

The hypothesis that temperature inversions partially drive the observed $NO_3$ reactivity within the canopy is reinforced by close inspection of the temperature profiles in period B in the inset to Fig. 2. At 18:00 UTC, no gradient in temperature (16.5 °C) between 5 and 41 m was observed. At 20:30 UTC, a positive gradient in temperature was observed at heights > 20 m, becoming

more distinct at 22:00 UTC. Under these conditions, vertical mixing from ground level to above-canopy-levels (ca. 20 m) is suppressed, whereas some mixing may still take place between 5 and 13 m where no temperature difference was observed. Weak (but non-zero) vertical mixing at the lower levels may be the cause of the high variability in $k_{night}^{VOC}$(5.5 m) whereby instabilities in the stratification at lower levels of the NBL allow sampling (at 5.5 m) of air with different (variable) time spent at lower/higher levels and thus with highly variable reactivity. The resulting high variability in $NO_3$ mixing ratios has been

documented for $NO_3$ measurements made close to the ground (Brown et al., 2003; Crowley et al., 2010).

In forests, the most abundant BVOCs are typically isoprene and monoterpenes (Hakola et al., 2012; Vermeuel et al., 2023). Since the corresponding rate coefficients for the reaction of $NO_3$ with monoterpenes are up to two orders of magnitude larger than that of isoprene (IUPAC, 2024), monoterpenes are expected to be the main contributor to $k^{VOC}$ during the ACROSS campaign. Relative monoterpene emission factors are temperature-dependent and described by $\exp(\beta(T\text{-}279\ K))$ with $\beta = 0.1$

$K^{-1}$ in forested environments (Guenther et al., 1993), resulting in a strong seasonal variation (Hakola et al., 2006; Vermeuel et al., 2023). Figure 3a shows that, during the day (green data points), with temperatures varying from 297 to 311 K the expected factor of 4 increase in the emission rate over this range is much larger than the observed change in $k^{VOC}$(5.5 m). This is however expected as the daytime concentrations of monoterpenes (MTs) will be determined not only by emission rates but also by their lifetime, which, in the clearing, will be reduced by reactions with daytime oxidants such as the OH and $O_3$ and

(possibly more importantly) transport out of the canopy (Bohn, 2006). At nighttime (blue and orange dots), there is no clear correlation between $k^{VOC}$(5.5 m) and temperature, although the highest values of $k^{VOC}$(5.5 m) are generally observed at lower temperatures. A plot of $k_{night}^{VOC}$(5.5 m) versus $\Delta T$ and coloured according to the temperature at 5 m (Fig. 3b) reveals that higher $NO_3$ reactivity is accompanied by large (negative) values of $\Delta T$. The maximum value of $k_{night}^{VOC}$(5.5 m) $\approx$ 2 $s^{-1}$ was observed when a strong temperature inversion ($\Delta T < -6$ K) coincided with a high nocturnal air temperature. During cooler nights (T(5

m) < 12 °C), $k^{VOC}$(5.5 m) was < 0.5 $s^{-1}$ even during periods with very strong temperature inversions. Our analysis thus shows that while temperature is an important factor influencing $NO_3$ reactivity at 5.5 m through enhanced rates of emission of BVOCs, the effect (at least at sub-canopy levels) is greatly amplified by temperature inversions which favour accumulation of MTs in



a shallow boundary layer close to the surface. In an upcoming publication we use monoterpene measurements to analyse $NO_3$ (and $N_2O_5$) mixing ratios and lifetimes at both 5.5 and 40 m heights during ACROSS and draw comparison between directly

measured $NO_3$ reactivity and that attributed to various BVOC.

**3.2 Nitrate radical lifetime within and above the canopy**

For the last five days of the ACROSS campaign (July 18 – July 23), the $NO_3$ reactivity was measured at a height of 40 m every even hour, with data obtained at 5.5 m every odd hour. The corresponding time-series of this measurement with a discussion of the vertical gradients observed during this period is appended in the Supplement (S2). With the data from this period, we

could derive the total (chemical) loss rate coefficient of $NO_3$ not only at 5.5 m but also at 40 m, which is given by

$$L^{NO_3} \approx k^{VOC} + J^{NO_3} + k_5[NO] \qquad (1)$$

where $k^{VOC}$ is our directly measured VOC-induced loss, $J^{NO_3}$ is the $NO_3$ photolysis loss rate coefficient (Fig. 1, panel a) and $k_5$ is the rate coefficient (IUPAC, 2024) for the reaction between NO and $NO_3$. Equation (1) neglects $NO_3$ loss processes resulting from direct (and indirect) heterogeneous reactions of $NO_3$ (and $N_2O_5$) as well as reactions with $HO_x$ and organic

radicals, which, for the ACROSS environment, is justified in the Supplement (S3). The $NO_3$ lifetime $\tau^{NO_3}$ is the inverse of the total $NO_3$ loss rate coefficient:

$$\tau^{NO_3} = 1/{L^{NO_3}} \qquad (2)$$

We plot $\tau^{NO_3}$ through the diel cycle (5.5 and 40 m) in Fig. 4. For calculating $L^{NO_3}$(5.5 m), the whole campaign period (as in Fig. 1) was considered since data gaps in both NO and $J^{NO_3}$ close to the ground and on top of the tower reduced the availability

of quasi-simultaneous values of $L^{NO_3}$(5.5 m) and $L^{NO_3}$(40 m) (see Supplement S4). At a height of 5.5 m, the nocturnal lifetime of $NO_3$ is 2–3 s and (counterintuitively) somewhat longer (4–5 s) during the day despite the photolysis of $NO_3$. At a height of 40 m, above the canopy, the daytime lifetime of $NO_3$ is close to 3–4 s, whereas at night it increases to 10–12 s. The similar lifetimes at both heights during the day imply a vertically well-mixed layer. On the other hand, the longer nocturnal lifetime at 40 m compared to 5.5 m can be attributed to decoupling from direct (ground-level and within canopy) sources of NO and

BVOCs (see section 3.1). The longer nocturnal lifetime of $NO_3$ at 40 m enabled it to be detected on some nights of the campaign, whereas $NO_3$ measurements were always below LOD close to the ground. A detailed analysis of the $NO_3$ (and $N_2O_5$) measurements will be presented in a future publication.

**3.3 Fate of the nitrate radical within the canopy**

A mean diel cycle of $k^{VOC}$(5.5 m) for the whole campaign is depicted in Fig. 5. The higher mean nighttime value ($k^{VOC}_{night}$(5.5

m) = $(0.24 \pm 0.27)$ s$^{-1}$, $k^{VOC}_{day}$(5.5 m) = $(0.12 \pm 0.04)$ s$^{-1}$ is a result of accumulation of BVOCs in a shallow, sub canopy layer with reduced rates of canopy-venting owing to temperature inversions. The observation of a daytime minimum and nighttime maximum in $k^{VOC}$(5.5 m) is consistent with measurements in the boreal forest in Finland (Liebmann et al., 2018a) where



terpene emissions dominated the fate of $NO_3$ and strong temperature-inversions were present at night. Values of $k^{VOC}$ in the boreal forest in autumn were a factor of 2–3 lower than those we measured in the temperate forest during ACROSS, which is presumably related to the lower temperatures as well as other factors such as leaf area index, vegetation-type and availability of oxidizing agents which affect the abundance of monoterpenes.

The fractional contribution $F^{VOC}$, of the reaction of $NO_3$ with VOCs to its total loss rate coefficient $L^{NO_3}$, is given by Eq. (3)

$$F^{VOC} = k^{VOC}/L^{NO_3} \tag{3}$$

and shown in Fig. 5. At night, $F^{VOC}$ (grey shaded area, blue line) is ca. 0.7–0.8, with the remaining 20 to 30 % assigned to reaction with NO. During the day, while photolysis (20 %) and reaction with NO (30 %) gain in importance, VOCs still account for 50 % of the $NO_3$ reactivity. The large daytime contribution of $k^{VOC}$ is partly related to the fact that actinic flux (and thus the rate of photolysis of $NO_3$) within the clearing at 5.5 m height is reduced compared to above canopy levels (Fig. 1, panel a). In addition, the photolysis frequency for $NO_2$ is also reduced, so that the distribution of $NO_X$ between NO and $NO_2$ is shifted towards $NO_2$ and away from NO, resulting in a reduction in $k_5[NO]$. Low daytime $NO_3$ photolysis frequencies have been reported by Decker et al. (2021) to result in $NO_3$ serving as the major VOC oxidant in wildfire plumes. We note at this point, that the photolysis frequencies measured in the clearing will be substantially larger than in the non-cleared forest.

By comparing $J^{NO_3}(5.5$ m$)$ with values observed above the canopy in the early morning and afternoon when the integrating-dome of the spectral-radiometer experienced only diffuse sunlight, we observe a reduction in $J^{NO_3}$ (5.5 m) by a factor > 10 (Supplement S5). A substantial reduction in in-forest actinic flux compared to in a clearing and above the canopy has been reported by Bohn (2006) for a temperate, deciduous forest, albeit of different tree type and density of foliage. As discussed by Bohn (2006), a reduction in $NO_3$ photolysis frequency implies that the relative importance of $NO_3$ (compared to OH and $O_3$) as a daytime oxidant in the forest canopy is even larger than that (~50 %) derived above. A caveat to this is that the formation of $NO_3$ requires the presence of $O_3$ and $NO_2$, both of which are likely to have substantial deposition terms in dense foliage in the non-cleared forest. We also recognise that, during the daytime, the in-canopy chemical-lifetimes of BVOCS may be much longer (see section 3.6) than the average residence time with respect to canopy-outflow, so that most BVOC daytime oxidation may take place above the forest.

## 3.4 Fate of the nitrate radical above the canopy

In Fig. 6, we plot the diel cycle of $k^{VOC}(40$ m$)$ which shows a daytime maximum and nighttime minimum, which is the opposite of that measured at 5.5 m (see Fig. 5) but is typical for measurement sites that are decoupled from near-ground emissions during the night (Crowley et al., 2011; Liebmann et al., 2018b; Dewald et al., 2022). The daytime mean value of $k_{day}^{VOC}(40$ m$)$ = $(0.07 \pm 0.02)$ s$^{-1}$ is similar to those measured at 5.5 m, which presumably results from vertical mixing that is rapid compared to chemical lifetimes of BVOCs and NO. In contrast, the average value of $k_{night}^{VOC}(40$ m$)$ = $(0.016 \pm 0.007)$ s$^{-1}$ is approximately one order of magnitude lower than $k_{night}^{VOC}(5.5$ m$)$. Recall however, that due its poor data coverage during the 5-day-period, $k^{VOC}(5.5$ m$)$ from the whole campaign is used for this comparison.



285    The daytime, chemical, gas-phase $NO_3$ loss processes at 40 m (see pie chart in Fig. 6) also differ significantly from those at

5.5 m (see Fig. 5) with the contribution of BVOCs almost halved (25.5 %) in favour of both NO (48 %) and photolysis (26.5

%). Reaction with NO is thus the main daytime loss process for $NO_3$ at 40 m, which is a result of greater NO mixing ratios at

this height during the 5-day-period. But even at night, NO mixing ratios during this period were still high enough (40-150

pptv) to compete with VOCs. In addition, some days between July 18 and July 23 were cloudy, which is why $NO_3$ loss via

290    reaction with NO prevails even at a height of 40 m. A time series comparing $J^{NO_3}$, $k_5[NO]$ and $L^{NO_3}$ at both heights during this

period is appended in the Supplement (S4).

**3.5 Vertical gradient (0-24 m) in $NO_3$ reactivity**

During the night from July 17 to July 18 (a night with a strong temperature inversion with $\Delta T$ between –4 and –6 K), gas-

phase $NO_3$ reactivity (resulting from reaction with both BVOCs and NO), $NO_2$ and $O_3$ were measured at heights between 0

295    and 24 m above ground level in 4 m steps. Measurements of $NO_2$ and $O_3$ at 40 m during the same night are also plotted in Fig.

2 (period P). As NO mixing ratios were not available at all heights, the $NO_3$ reactivity was not corrected for this and thus

includes $NO_3$ removal via reaction with NO, i.e. $k^{VOC}$ becomes $k^{VOC+NO}$.

Between 20:00 and 00:00 UTC five profiles of $k^{VOC+NO}$ were measured, each taking < 20 min. The first profile was measured

after an increase in $k^{VOC}_{night}$(5.5 m) which was associated with the onset of the temperature inversion. $k^{VOC+NO}$ (averaged over

the 5 single profiles) is plotted versus height in Fig. 7c along with the data obtained at 40 m in the following nights (see section

3.2). The data reveal a strong trend in $NO_3$ reactivity with the highest values (0.34 $s^{-1}$) measured at ground level, where NO is

expected to have a greater impact, gradually decreasing with height to a value of ca. 0.08 $s^{-1}$ at 24 m. The data at 40 m

(uncorrected for NO) are broadly consistent with this trend and, together with Fig. 6, imply that NO is the main contributor to

$NO_3$ reactivity above canopy level. The gradient in $NO_2$ (Fig. 7b) shows no clear trend for heights below 24 m and is determined

by its nighttime production (mainly the reaction between NO and $O_3$) which depends on the availability of NO and the

availability of $O_3$ (which has a positive gradient). As modelled in Stutz et al. (2004), a negative nocturnal gradient in NO with

height is expected, notably due to low level soil emission and reaction with $O_3$ (Andersen et al., 2024). In addition, $NO_2$ loss

via deposition is expected to be more important at the lower levels. These processes appear to roughly counterbalance each

other on this night, resulting in an almost constant mixing ratio between ground level and 24 m. This observation is consistent

with those in Stutz et al. (2004), who could, if at all, only find very weak gradients. Fig. 7a also includes the vertical profiles

of $O_3$ which increases from ca. 27 ppbv at ground-level to 58 ppbv at 40 m presumably a result of near-ground loss processes

such as deposition and reaction with NO / BVOCs and lack of entrainment of $O_3$ from above the canopy, where higher mixing

ratios were measured (Fig. 1, panel c) (Brown et al., 2007a). The measured vertical profiles of $O_3$ were much less distinct in

Stutz et al. (2004), which contradicted their model calculations that suggested a monotonic increase within the first 100 m,

broadly consistent with our case study.





### 3.6 Fractional contribution of NO₃ to BVOC oxidation below the canopy

Our results show that, within the canopy, the nitrate radical is lost by reactions with BVOCs not only during the night but also during the day. Here we calculate the fractional contribution of $NO_3$ to the oxidation of a dominant BVOC, α-pinene (Guenther et al., 2012; Liebmann et al., 2018a; Vermeuel et al., 2023). As detailed in Eq. (4) the overall oxidation rate coefficient for α-

pinene ($L^{\alpha\text{-pinene}}$) can be calculated from the concentrations of each oxidant (OH, $NO_3$ and $O_3$) and the corresponding rate coefficients $k_{x+\alpha\text{-pin}}$ of $5.3 \times 10^{-11}$, $6.2 \times 10^{-12}$ and $9.6 \times 10^{-17}$ cm³ molecule⁻¹ s⁻¹ for the reaction of OH, $NO_3$ and $O_3$ with α-pinene at 298 K, respectively (IUPAC, 2024).

$$L^{\alpha-\text{pinene}} = k_{\text{OH}+\alpha-\text{pinene}}[\text{OH}] + k_{\text{O}_3+\alpha-\text{pinene}}[\text{O}_3] + k_{\text{NO}_3+\alpha-\text{pinene}}[\text{NO}_3]_{\text{ss}} \qquad (4)$$

As measured $NO_3$ mixing ratios near ground-level were below the LOD of 2 pptv throughout the whole campaign, we have

derived steady-state mixing ratios, $[NO_3]_{ss}$, from the ratio of production ($k_2[NO_2][O_3]$) and overall loss rates $L^{NO_3}$ (Eq. 1) (Heintz et al., 1996; Brown et al., 2003; McLaren et al., 2010; Crowley et al., 2011). Note that the OH measurements were carried out with a chemical ionization mass spectrometer (CIMS, Kukui et al. (2008), see Supplement S3) ca. 17 m apart from the MPIC container. Figure 8 (upper panel) depicts the campaign-averaged, median mixing ratios of OH, $O_3$ and $NO_3$ (steady-state) throughout the diel cycle. Nighttime, steady-state $NO_3$ mixing ratios were 0.1-0.2 pptv during late evening until midnight,

decreasing to < 0.1 ppt between midnight and dawn, which is related to the reduced availability of $O_3$. Remarkably, daytime $NO_3$ values were similar and occasionally even higher than nighttime values, emphasising the fact that photolysis was not the major daytime loss process for $NO_3$. As expected, OH mixing ratios were highest during the daytime (peak value of 0.14 pptv), but were also present at night (0.01–0.03 pptv). Nighttime OH is formed in the oxidation of terpenes by $O_3$ and in secondary reactions of $RO_2$ and $HO_2$ (formed in same process) with NO reactions. Median $O_3$ mixing ratios reached ca. 44 ppbv during

the day and continuously decreased to ca. 15 ppbv during the night and the early morning.

The fractional contribution of each oxidant to $L^{\alpha-\text{pinene}}$ is presented as a median diel profile for the whole campaign in the lower panel of Fig. 8. Based on the above analysis, in the summertime forested environment probed during ACROSS, $NO_3$ contributed *only* ca. 17 % to the nighttime oxidation of α-pinene, a value that is exceeded by both OH (ca. 24 %) and $O_3$ (60 %). The daytime dominance of OH and $O_3$ (on average ca. 50 and 41.5 %, respectively) is expected, whilst, with a contribution

of 8.5 % the diel- and campaign-averaged contribution of $NO_3$ is still significant, which is in agreement with recent publications (Schulze et al., 2017; Liebmann et al., 2018b; Mermet et al., 2021; Dewald et al., 2022) and model calculations showing that $NO_3$ oxidizes BVOCs below the canopy level during daytime in both coniferous and deciduous forests (Forkel et al., 2006; Fuentes et al., 2007). We further note that the oxidation of BVOCs by OH, $O_3$ and $NO_3$ results in greatly different products, so that in terms of formation of organic nitrates, $NO_3$-initiated oxidation can still dominate (Liebmann et al., 2019). A detailed

analysis on the role of $NO_3$-initated organic nitrate formation will be presented in a further publication describing the ACROSS campaign. Recall that the fractional contributions are highly dependent on the ratios between the rate coefficients of the oxidants and are thus different for other monoterpenes. As shown in the Supplement (S6), this is reflected in lower fractional



contributions of $NO_3$ to the oxidation of β-pinene and limonene with values of 12.4 and 13.5 % at night and 4.4 and 6.2 % during the day.

Our analysis is not consistent with the generally accepted text-book paradigm that OH-initiated oxidation processes are predominant during the day and $NO_3$-initiated oxidation prevails at night. However, an important caveat to our analysis is the neglect of daytime transport (venting) of BVOCs out of the forest. If this proceeds at time-scales that are short compared to the chemical lifetime of BVOCs, then in-canopy oxidation will be of reduced importance compared to oxidation once BVOCs have been transported to the free troposphere (Bohn, 2006).

## 355 4 Conclusions and summary

During a field campaign in a peri-urban, temperate (oak and pine) forest in France during a summer period in 2022, the reactivity of the $NO_3$ radical towards VOCs was measured both within the canopy (height of 5.5 m), above the canopy (height of 40 m) and on one night at several heights between 0 and 24 m. $NO_3$ lifetimes were generally short (1–3 s) which were driven mainly by the abundance of BVOCs. Diel cycles of $NO_3$ at 5.5 m and 40 m were distinct, with the highest reactivity at 40 m

occurring during the day and the highest reactivity at 5.5 m measured at nighttime. The highest nighttime reactivities were associated with high temperatures (driving the BVOC emissions) and with strong nighttime temperature inversions (preventing mixing of BVOCs and NO out of the nocturnal surface layer). At 5.5 m, BVOCs represented the dominant loss term for $NO_3$ both during the night (70–80 %) and during the day (~50 %), which is partially a result of reduced $NO_3$ (and $NO_2$) photolysis frequencies at sub-canopy heights. $NO_3$ reactivity decreased rapidly with height above the ground with nocturnal lifetimes

(with respect to reaction with VOCs) of > 100 s at 40 m and as low as 2.5 s at ground-level. This gradient is driven largely by BVOC and NO emissions into a shallow, stratified near-surface layer under canopy height.

The conventional wisdom, that OH is a daytime oxidant and $NO_3$ a nighttime oxidant appears not to apply to forested regions with significant BVOC emissions where both $NO_3$ and OH have important roles throughout the diel cycle. Hence, $NO_3$-initiated organic nitrate formation could become significant during the day, whereas OH-initiated nocturnal chemistry would be

enhanced in such environments.

*Data Availability.* All data can be found on https://across.aeris-data.fr/catalogue/.

*Author contributions.* PD analysed the data and wrote the original draft of the manuscript and, together with JNC, revised it.
CC and VM were responsible for the campaign organization with contributions from individual group leads. All authors provided measurements and commented on the manuscript.

*Competing Interests.* At least one of the (co-)authors is a member of the editorial board of Atmospheric Chemistry and Physics.



*Acknowledgements.*

JC acknowledges Chemours for the provision of a FEPD-121 sample used to coat the flowtube and the Deutsche Forschungsgemeinschaft (project "MONOTONS", project number: 522970430). STA thanks the Alexander von Humboldt foundation for funding her stay at the MPIC. The ACROSS project has received funding from the French National Research Agency (ANR) under the investment program integrated into France 2030, with the reference ANR-17-MPGA-0002, and it

was supported by the French National program LEFE (Les Enveloppes Fluides et l'Environnement) of the CNRS/INSU (Centre National de la Recherche Scientifique/Institut National des Sciences de l'Univers). IMT Nord Europe acknowledges financial support from the CaPPA project, which is funded by the French National Research Agency (ANR) through the PIA (Programme d'Investissement d'Avenir) under contract ANR-11-LABX-0005-01, the Regional Council "Hauts-de-France" and the European Regional Development Fund (ERDF). Data from the ACROSS campaign are hosted by the French national

centre for Atmospheric data and services AERIS. AK acknowledges financial support from the VOLTAIRE project (ANR-10-LABX-100-01) funded by ANR through the PIA (Programme d'Investissement d'Avenir).

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



**Figures**



**Figure 1: Time series of NO₃ photolysis rates ($J^{NO_3}$, panel a), temperature (T, panel b), ozone (O₃, panel c), nitrogen dioxide (NO₂, panel d), nitric oxide (NO, panel e) and VOC-induced NO₃ reactivity ($k^{VOC}$, panel f) sampled at 5.5 m (orange, 3.2 m and 5 m for**
**NO and T, respectively) and 40 m (blue) above ground level. Major and minor ticks on the x-axis represent 00:00 UTC of the corresponding date. Dashed lines separate periods with air of Atlantic and continental origin. Nighttime periods are grey-shaded.**



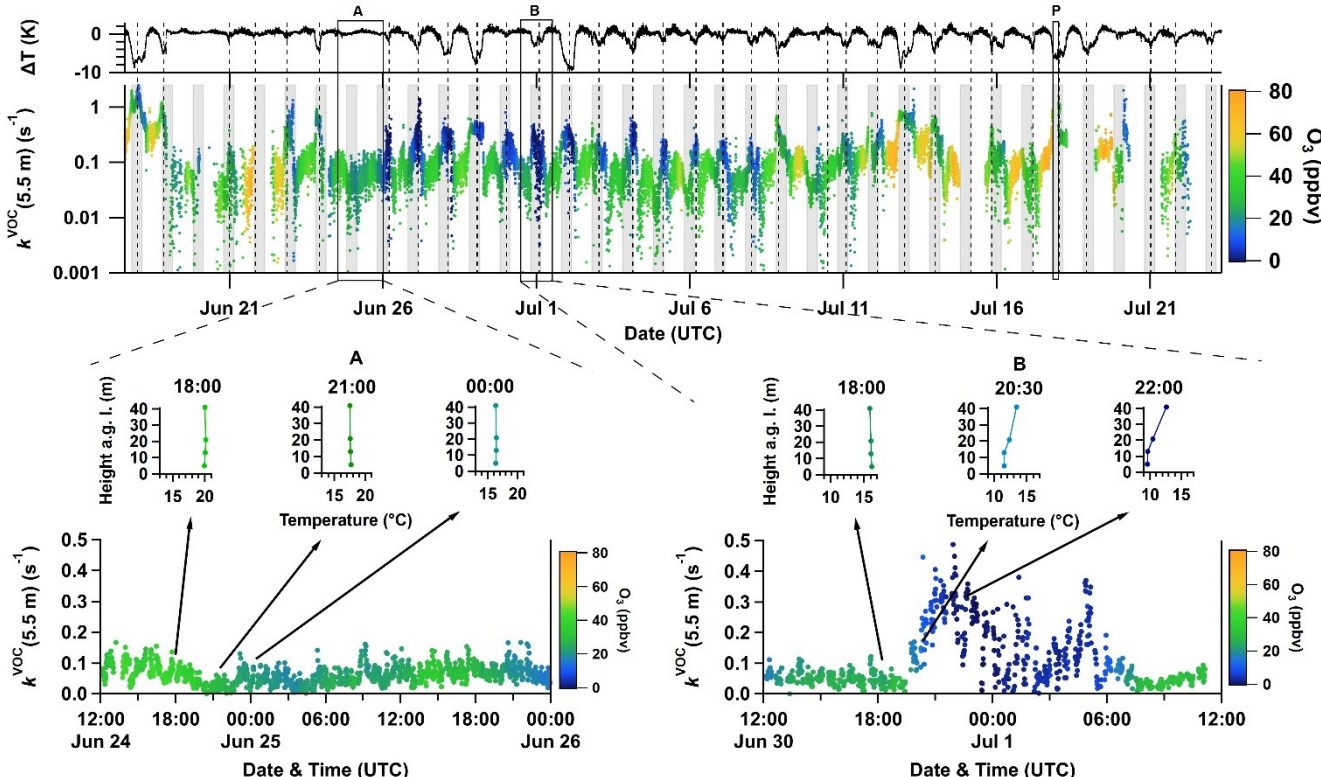

**Figure 2: Time series of $k^{VOC}$ (second panel, logarithmic scale) near ground level (5.5 m) coloured by O₃ mixing ratios measured at the same height (colour scale on the right). Nighttime periods are grey-shaded. The difference between temperatures measured at 5 m and 41 m (ΔT) is plotted in the first panel. Temperature inversions are marked by dashed, vertical lines. Periods A (lower left panel) and B (lower right panel) exemplify daytime-nighttime transitions both with (right) and without (left) clear temperature inversions, with NO₃ reactivity plotted along with temperature profiles at selected times. A vertical profile of $k^{VOC}$ was measured in period P (see section 3.5).**



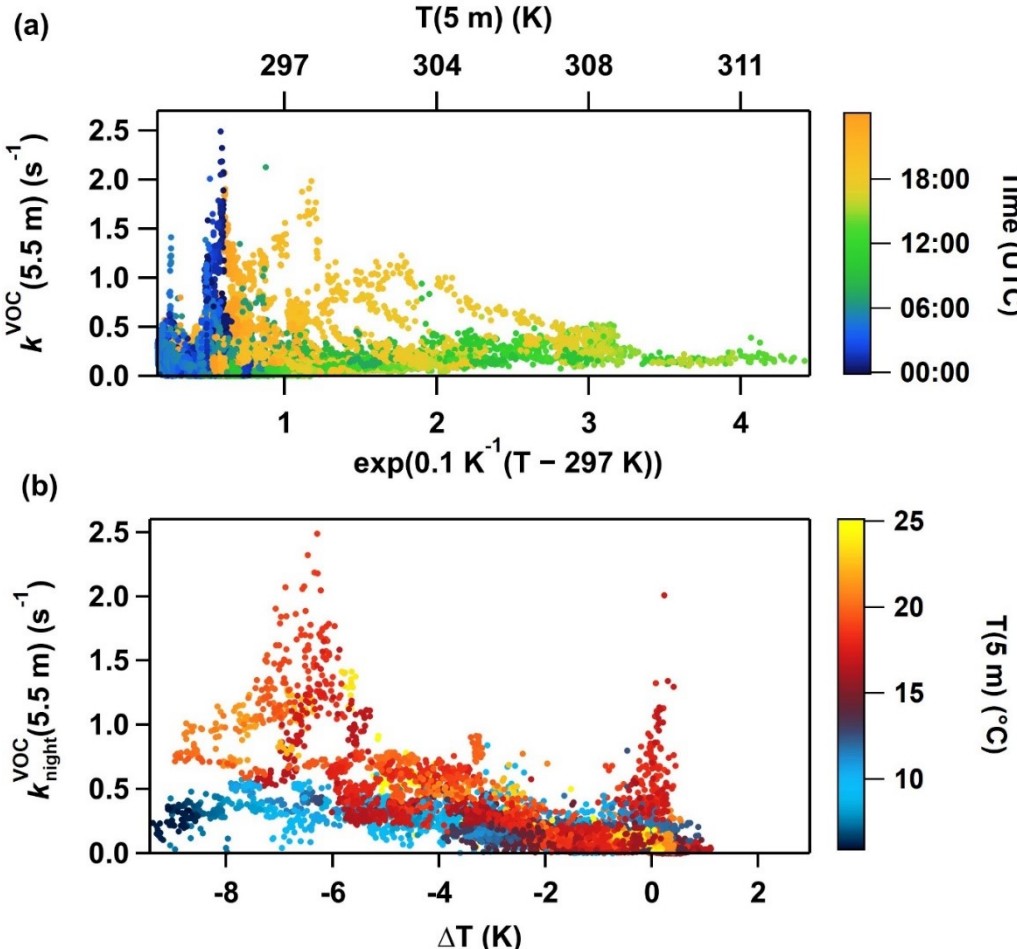

**Figure 3: (a)** NO₃ reactivity at 5.5 m ($k^{VOC}(5.5\,m)$) plotted against the calculated change in the monoterpene emission factor ($E_{MT}$) relative to 297 K using $E_{MT} \propto \exp(0.1K^{-1}(T-297K))$ (Guenther et al., 1993). Data points are coloured according to time of the day (UTC) with night/morning (blue), daytime (green) and afternoon/night (orange). **(b)** Nocturnal ground-level NO₃ reactivity $k^{VOC}_{night}(5.5\,m)$ plotted versus the temperature difference ($\Delta T$) between 41 m and 5 m. The data points are coloured according to the ground-level temperature T(5 m).



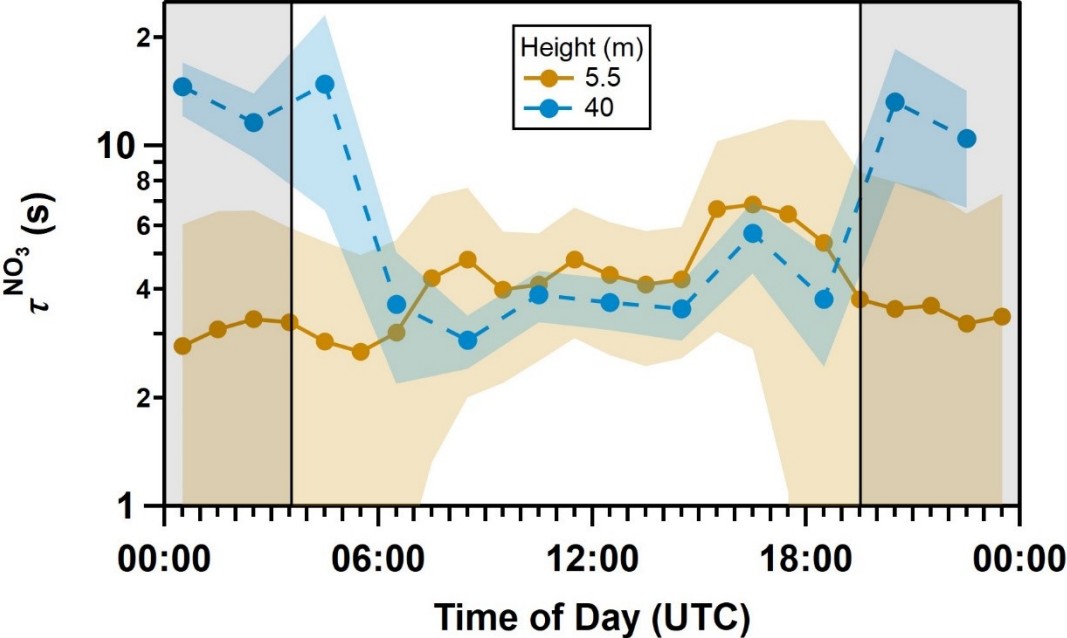

Figure 4: Mean diel cycle of overall $NO_3$ lifetime ($\tau^{NO_3}$) calculated from Eq. (2) at 5.5 m (dark orange points with solid line, June 17 – July 23) and 40 m (blue points with dashed line, every 2h, July 18 – July 23). Shaded areas represent the standard deviation (1σ). The nighttime period (19:30 to 03:30 UTC) is shaded grey.



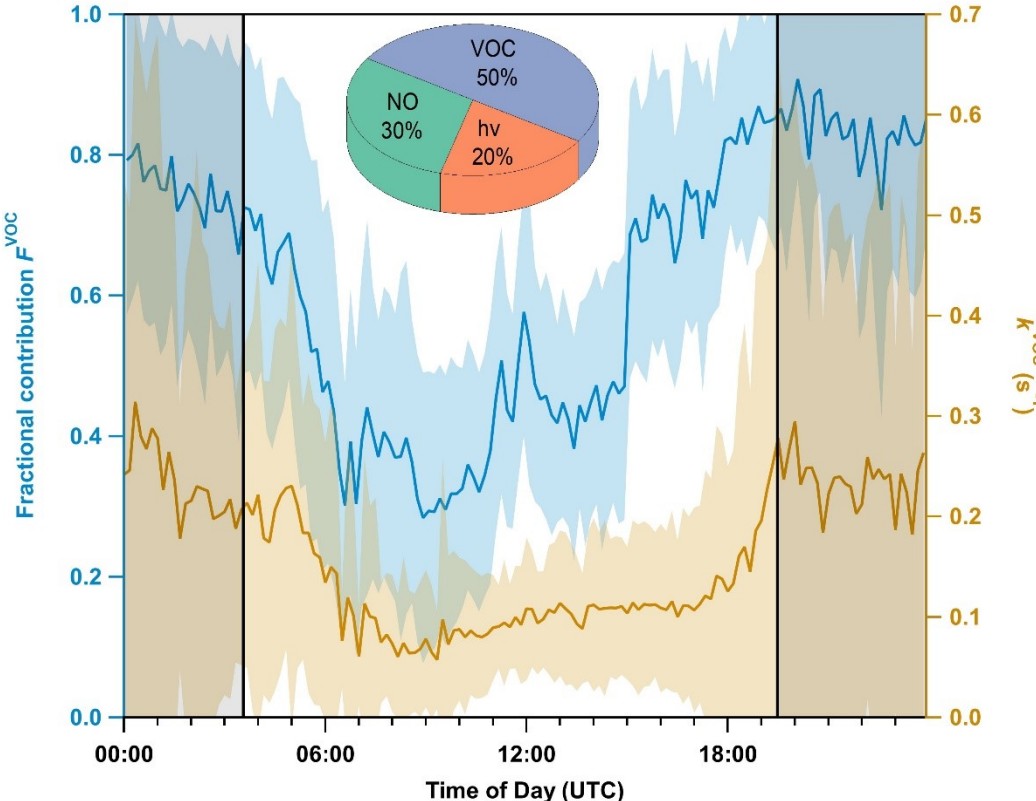


**Figure 5: Mean diel profiles (10 min averages, June 17 – July 23) of VOC-induced NO₃ reactivity ($k^{VOC}$, right y-axis) at 5.5 m along with its fractional contribution ($F^{VOC}$, left y-axis) to the overall NO₃ loss. Shaded areas represent the standard deviation (1σ). The nighttime period is grey-shaded and separated by black solid lines. The pie chart shows the fractional contribution of each process in Eq. (1) to the overall NO₃ loss term during the day.**





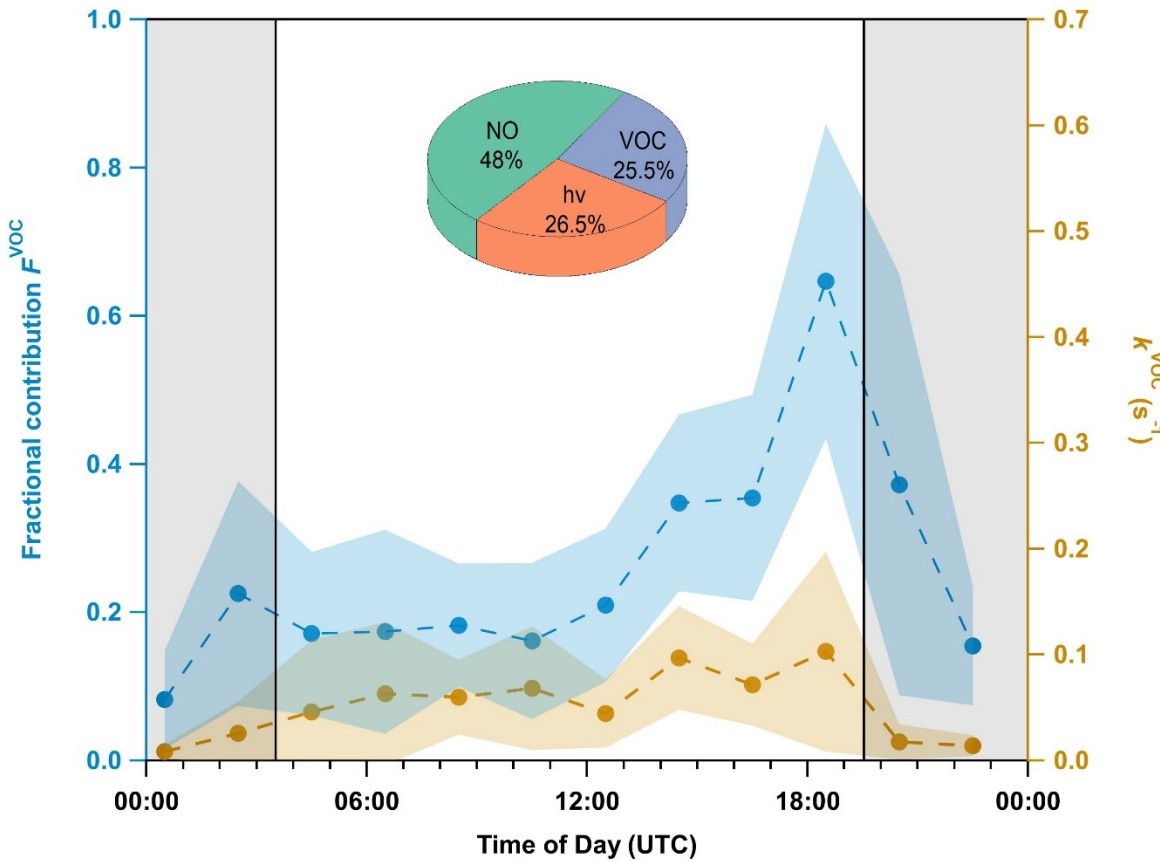


**Figure 6: Mean diel profiles (1h averages, every two hours, July 18 – July 23) of VOC-induced NO₃ reactivity ($k^{VOC}$) at 40 m along with its fractional contribution ($F^{VOC}$, blue line) to the overall NO₃ loss. Shaded areas represent the standard deviation (1σ). The nighttime period is grey-shaded. The pie chart shows the fractional contribution of each process in Eq. (1) to the overall NO₃ loss term during the day.**




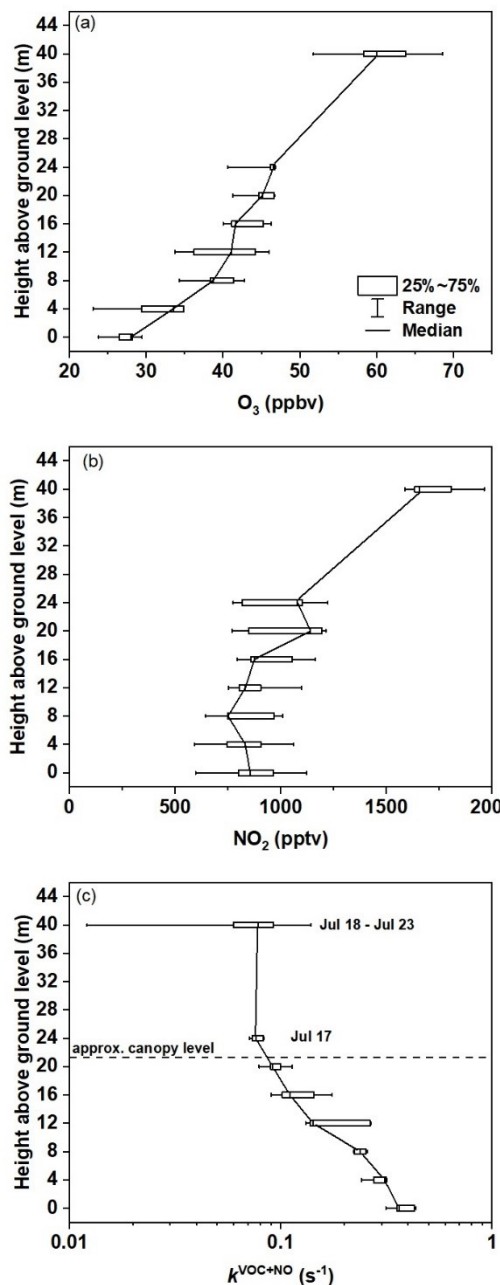


**Figure 7: Box-and-whisker plots (full range, 25th, 50th and 75th percentiles) of vertical profiles of (a) O₃ (b) NO₂ and (c) total gas-phase NO₃ reactivity ($k^{VOC+NO}$) measured during the temperature-inverted night of Jul 17 to Jul 18 between 20:00 and 00:00 UTC. NO₃ reactivities measured on top of the tower (40 m) during the nights between July 18 to July 23 are also shown.**



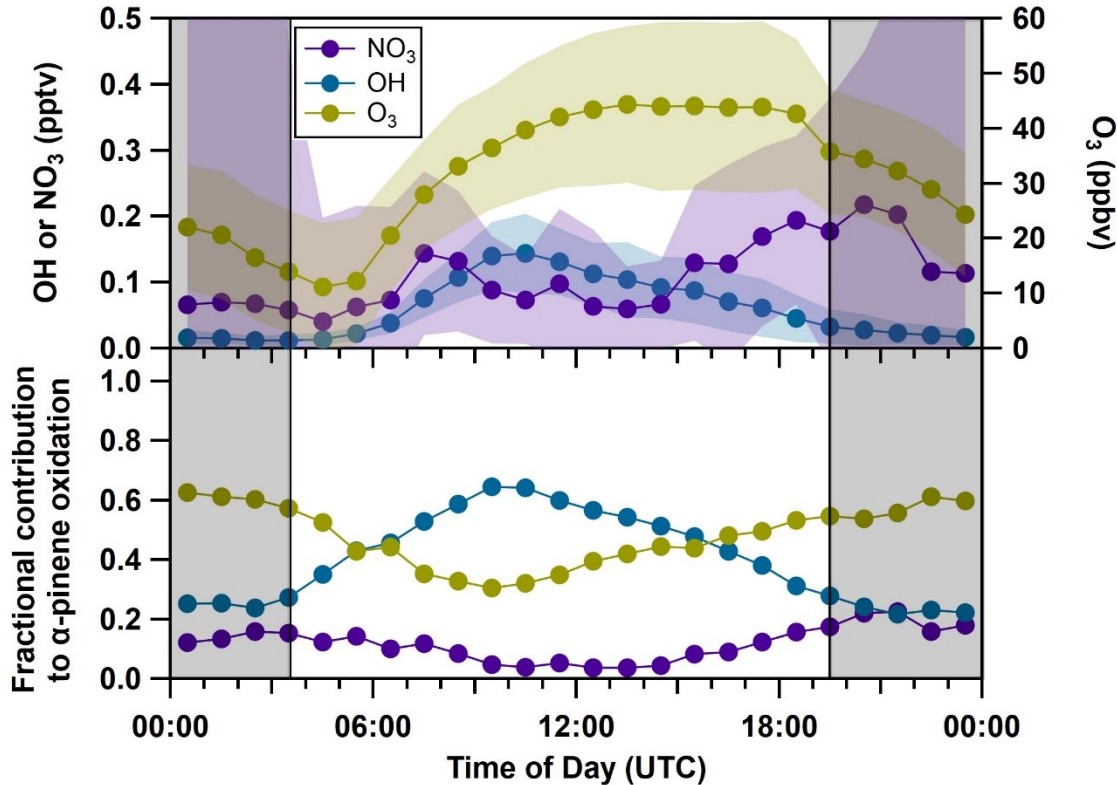

**Figure 8: Campaign-averaged median (circles) diel profiles (1h) of the oxidants (upper panel) [NO₃]ₛₛ (violet), OH (blue) and O₃ (dark yellow) and their contribution to the overall oxidative loss of α-pinene according to Eq. (4) close to the ground (lower panel). The nighttime period is grey-shaded. Shaded areas represent the standard deviation (1σ).**