# Peer review of "NO3 reactivity during a summer period in a temperate forest below and above the canopy"

_EGUsphere, 2024_

## Referee Comment (RC1)

General Comments:

The work by Patrick Dewald et al. is relevant to the atmospheric science community, well organized and well written. The quantitative details regarding $NO_3$ reactivity and fractional contribution to biogenic volatile organic compound (BVOC) oxidation appear robustly determined, clear and generally consistent with literature. Their results suggest an important conclusion that $NO_3$ can play a significant role during daytime oxidation which is counter to textbook atmospheric chemistry but not necessarily unheard of. However, there are a couple of points of concern.

First, the authors are reserving a more in-depth analysis of $NO_3$ oxidation from the same measurements and campaign for a future publication. These details seem highly relevant to information presented here and I question why they are better suited for an independent publication. Doing so removes context which would define the significance of the results. Second, it is unclear if the cause for significant daytime oxidative contribution from $NO_3$ is due to reduced photolysis or increased VOC concentration. The answer to this point may be answered in the author's future publication but, again, it appears highly relevant here.

Specific Comments in order of line number:

1) Lines 100–108: Here, the authors describe the inlets however I did not see any discussion of inlet loss rates or the potential effects on reactivity measurements. I am concerned that the 40 m tall measurements (with a 20 m length and ~ 5 s residence time) may have wall loss of VOCs. Such an effect would result in reduced $NO_3$ reactivity as seen in Figure 1. Some mention of this point would strengthen the validity of these measurements.

2) Section 3.1: the overall discussion here is clear, but I am left with a question of how the temperature inversion effects the production of $NO_3$, which will contextualize the importance of the resulting $k^{VOC}$. As stated by the authors, the temperature inversions are associated with decreased $O_3$ (and presumed decrease in $NO_3$ production) but increased $k^{VOC}$. The effect of these two counteracting variables is absent from the discussion. I believe such a discussion is needed.

3) Lines 220–222: The authors state that their analysis (Figure 3) suggests temperature is an important factor that influences $NO_3$ reactivity by BVOC emission. However, I do not find the same conclusion because such a relationship is not apparent in Figure 3. It is clear that the BVOC emission is dependent on temperature (by design) but, as the authors state in the discussion, the relationship between emission and $NO_3$ reactivity is clouded by competing oxidants during the daytime. The conclusion from this discussion appears to be simply that inversion has a strong effect (Figure 3b). I suggest the authors offer a better justification for their conclusion, remove this point, or provide additional VOC data to elaborate.

4) Lines 264–265: The point by the authors that Decker et al. (I am the first author of this cited paper) find $NO_3$ is a major oxidant during daytime for wildfire plumes due to their reduced photolysis rates is incorrect. Instead, the cited work concludes that $NO_3$ is a major daytime oxidant in wildfire plumes because smoke plumes are a huge source of reactive VOCs which outcompetes photolysis. This point is (admittedly hard to find) most clearly argued for a smoke plume (Castle) with a $jNO_3$ of 0.14 $s^{-1}$ sampled mid-day which is comparable to mid-day

photolysis rates presented here (Figure 1). In that cited case, photolysis accounted for ~0.6 % of $NO_3$ loss but the large source of VOCs accounts for the remainder of $NO_3$ loss. The overall conclusion of the cited work is also that $NO_3$ oxidation can be significant during daytime (in the case of biomass burning plumes).

This bring up the question of whether the large daytime $F^{VOC}$ here is the result of reduced photolysis or large BVOC concentrations (or both). As written, the authors appear to suggest that large daytime $F^{VOC}$ is due to reduced sunlight although a definitive conclusion is not clear. The authors acknowledge that $NO_3$ production will also be affected by reduced sunlight, but the effect is not discussed.

A major conclusion of this work is that $NO_3$ is a significant daytime oxidant in this sub-urban forest yet a reason is not clear. Such a conclusion is counter-intuitive based on textbook atmospheric chemistry (as mentioned by the authors). As such, understanding the conditions that cause such a result is highly relevant and important to the atmospheric community. I feel strongly that an interpretation of the cause of a large daytime $F^{VOC}$ would provide highly important context for the author's conclusion that $NO_3$ reactivity is significant during the daytime. I strongly suggest the authors provide further analysis (e.g. $NO_3$ production, information on BVOCs and concentrations) so that the importance of this result is better realized.

5) Section 3.6: I have a minor point that (to my understanding) equation 4 is a reactivity, not an oxidation rate, because it is first order and does not include the VOC concentration. Perhaps it could also be called the total loss rate as was done for $NO_3$.

Technical Details:

Line 283: "Recall however, that due its poor" typo here.

Line 296: "Figure 2 (period p)" seems wrong.

Figure 3(a): I suggest to change to x-axis label to something more interpretable by the reader such as "Calculated BVOC emission (unitless?)".

Figure color scales: I encourage the authors to use color scales that are readable in grey-scale. This improves accessibility to color blind readers. For example, see Crameri et al. 2020 (https://doi.org/10.1038/s41467-020-19160-7).

Figure markers: to the above point I encourage the authors to use opposing marker styles (such as filled and empty circles) so that the reader does not rely on color to differentiate these markers.

---

## Author Comment (AC1)

**Reply to RC2**

*In the following, the referee's comments are reproduced (black) along with our replies (blue) and changes made to the text (red) in the revised manuscript. Line numbers refer to those in the initial submission.*

**General comments**

The paper by Dewald et al. describes a summer-season field study of $NO_3$ reactivities measured in a temperate forest in France. Measurements were made in a small clearing close to ground and on a 40 m tower. The results are discussed in terms of diurnal variations of contributions from VOCs, NO and photolysis to the overall $NO_3$ reactivity, with VOC reactivities found to be significant during night and day. Based on measured and estimated concentrations of OH, $O_3$ and $NO_3$, the relative importance of these reactants for the oxidation of selected BVOCs is estimated, revealing a significant contribution of oxidation through $NO_3$, again during night and day, at least close to ground.

We thank the referee for taking the time to assess our manuscript and for the provision of helpful comments.

The paper is relevant, well written and structured and therefore suitable for publication in ACP. However, the work would gain relevance if information on the BVOC composition were available and if there is any missing $NO_3$ reactivity. In other words: how well is the $k^{VOC}$ understood in this environment?

The main objectives of this work are to identify factors that control the temporal and vertical variability in $NO_3$ reactivity and to assign the contribution of VOCs, NO and photolysis to $NO_3$ removal. A rigorous assignment of $k^{VOC}$ to specific compounds is not (yet) possible due to lack of speciated monoterpene measurements at ground level. Several of the authors of this publication plan to address this question in a further publication (including $NO_3$ measurements at 5.5 and 40 m), as soon as speciated VOC measurements at ground level are available. In the reply to reviewer 1, we outline why combining these papers is neither necessary nor conducive to readability.

Below are some comments that may be considered in a revised version of the paper.

**Specific comments**

Line 22: Given the strong variability and peak values of the $k^{VOC}$ at night, an arithmetic mean and a standard deviation are not useful to summarize the results. Consider using a range instead or percentiles.

We agree and now provide the 25th and 75th percentiles of $k^{VOC}_{night}$ and modified L22, L250 and L282 accordingly:

L22: At nighttime, mean values (and 25th-75th percentile ranges) of $k_{night}^{VOC}(5.5\,\text{m}) = (0.24_{-0.06}^{+0.32})$ s$^{-1}$ and $k_{night}^{VOC}(40\,\text{m}) = (0.016_{-0.007}^{+0.018})$ s$^{-1}$ indicate a significant vertical gradient […].

L250: The higher mean nighttime value ($k_{night}^{VOC}(5.5\,\text{m}) = (0.24_{-0.06}^{+0.32})$ s$^{-1}$ compared to $k_{day}^{VOC}(5.5\,\text{m}) = (0.12 \pm 0.04)$ s$^{-1}$) is a result of accumulation of BVOCs in a shallow, sub canopy layer with reduced rates of canopy-venting owing to temperature inversions.

L282: . In contrast, the average value ($\pm$ 25th and 75th percentile, respectively) of $k_{night}^{VOC}(40\,\text{m}) = (0.016_{-0.007}^{+0.018})$ s$^{-1}$ is approximately one order of magnitude lower than $k_{night}^{VOC}(5.5\,\text{m})$.

Line 28: The "total loss rate" $L^{NO_3}$ is later defined (Eq. 1) as the total NO$_3$ reactivity, i.e. a (pseudo first-order) loss rate coefficient. Loss and production rates are usually named $L$ and $P$ but defined as products of concentrations and rate constants resulting in units of cm$^{-3}$s$^{-1}$ (or ppb h$^{-1}$, dependent on context). $k^{TOT}$ would be a better choice (see technical comments).

We agree and changed $L^{NO_3}$ to $k^{tot}$ throughout the whole manuscript.

Line 30: Perhaps clarify that the conclusions regarding the contribution of NO$_3$ to $\alpha$-pinene degradation and the chemical lifetimes of BVOCs are based on measured OH and O$_3$ concentrations and estimated steady-state NO$_3$ concentrations.

We now write in L30:

Based on **measured** OH, O$_3$ and **calculated** NO$_3$ concentrations […].

Line 60: Both photolysis reactions recycle NO$_x$ but only (R7b) recycles NO$_2$. Consequently, (R7b) also regenerates O$_x$ (NO$_2$ + O$_3$) but (R7a) is a net daytime loss of O$_x$, another perhaps underestimated daytime effect of NO$_3$ chemistry.

Thank you for pointing this out, we rephrased the sentence in L60 accordingly:

While both the reaction with NO (R5) and photolysis (R7) recycle either NO or NO$_2$, […].

Given the slow rate coefficient of ca. 3 x 10$^{-17}$ cm$^3$ molecule$^{-1}$ s$^{-1}$ at 298 K (IUPAC, 2024) for the reaction between NO$_2$ and O$_3$, the impact of (R7b) on the recycling of NO$_3$ is expected to be very minor.

Line 120: Was the transmittance of BVOCs through the combination of sampling lines, filter, and glass flask tested? BVOCs with low volatility and high reactivity may get lost on the way to the flow tube.

The transmittance of BVOCs through this system was tested by switching between flask and bypass at 03:30 and 19:30 UTC and observing no change in k$^{VOC}$ levels. This indicates that the compound that significantly contribute to k$^{VOC}$ are not lost in the glass flask. Given the higher reactivity of uncoated borosilicate glass surfaces compared to PFA and the

shorter residence time in the tubing/filter, a significant transmission loss in the latter appears unlikely. We now clarify this in the manuscript by adding in L118-120:

From July 18, air was sampled through the glass flask throughout the diel cycle. No difference in $k^{VOC}$ was observed directly after switching between "daytime mode" (no flask) and "nighttime mode" (sampling through flask), indicating that no compounds significantly contributing to $k^{VOC}$ are lost in the glass flask.

In addition, the inlet filter was replaced every three days to ensure that aging of the filter does not lead to transmission loss. We thus further note in L106:

The filter was replaced every three days.

Line 132: Your correction subtracts the remaining NO reactivity from the measured reactivity to derive $k^{VOC}$. Under conditions with low $k^{VOC}$ and [NO]>[$O_3$] the uncertainties are probably greater than the stated average 26%. Does your numerical simulation and correction procedure provide realistic, condition dependent uncertainty estimates?

Yes, as detailed in Liebmann et al. (2017), the calculation of the uncertainties takes ambient levels of $k^{VOC}$, $NO_2$ and NO at each data point into account. A few data points of $k^{VOC}$ during the ACROSS campaign thus have an associated uncertainty > 50 %. The campaign-averaged value stated in the manuscript is supposed to serve as a ballpark value.

Line 154: "Photolysis rates" should be named photolysis rate coefficients or photolysis frequencies consistently throughout the text.

Done, we replaced "photolysis rates" with "photolysis frequencies" throughout the whole manuscript.

Line 157: The paper of Meusel et al., contains no information on $NO_3$ absorption cross sections. I assume you used IUPAC or NASA-JPL recommendations for quantum yields and cross section which should be cited here.

We used IUPAC- and NASA-recommended quantum yields and changed the reference in L157 accordingly:

Actinic fluxes were converted to photolysis frequencies of $NO_3$ and of other compounds using IUPAC- and NASA-evaluated absorption cross sections (Burkholder et al., 2016; IUPAC, 2024).

Line 246: A future publication on $NO_3$ measurements? In Sect. 3.6 it is stated that $NO_3$ mixing ratios were always below the LOD which is consistent with the steady-state estimates < 0.2 ppt in Fig. 8.

On some nights, $NO_3$ as well as $N_2O_5$ mixing ratios above LOD were detected on top of the tower and $N_2O_5$ was detected at ground level, which provides the basis for the above-mentioned future publication. We emphasized this point in L245:

**The longer nocturnal lifetime of NO₃ at 40 m enabled it to be detected on some nights of the campaign, whereas NO₃ measurements were, unlike N₂O₅, always below LOD close to the ground.** A detailed analysis of the NO₃ **(and N₂O₅)** measurements will be presented in a future publication.

**Technical comments**

Line 95: Specify where the wind measurements were made in the caption of the figure in the Supplement (5 m, 40 m?)

The wind measurements were carried out on top of the tower. We now mention this in the figure caption of Fig. S1.

Line 137: "$L^{NO_3}$ which is the loss term" maybe better: "$k^{TOT}$ which is the total reactivity"

Done, we replaced "$L^{NO_3}$" with "$k^{tot}$" throughout the whole manuscript.

Line 231, Eq. (1): $k^{TOT}$ would fit better throughout the text with the upper index reserved for reactants, i.e. $k^{TOT}$, $k^{VOC}$, $k^{NO}$ but $\tau_{NO3}$, $J_{NO3}$ with the lower index reserved for the target species if necessary. Define $k^{NO} = k_5[NO]$ and $J_{NO3} = k_{7a} + k_{7b}$

Correction made.

Line 297 ff: Better $k^{VOC} + k^{NO}$

Correction made.

Line 320 ff: Better "$k_{\alpha\text{-pinene}}$" than "$L^{\alpha\text{-pinene}}$"

Correction made.

Line 325: Better "…the ratio of production rates $k_2[NO_2][O_3]$ and overall loss rate coefficients $k^{TOT}$"

Correction made.

Figs. 4, 5, 6, 8: How was the nighttime period 03:30-19:30 determined? Local noontime is close to 12:00 UTC. The night ends before sunrise and starts before sunset (checked for July 10[th]).

The NO₃ photolysis rates as derived from actinic flux measurement on top of the tower served as measure for the estimation periods that were unaffected by sunlight (see e.g. Fig. S5).

**References**

Burkholder, J. B., Sander, S. P., Abbatt, J. , Barker, J. R, Huie, R. E., Kolb, C. E., Kurylo, M. J., Orkin, V. L., Wilmouth, D. M.., and Wine, P. H.: Chemical Kinetics and Photochemical Data for Use in Atmospheric Studies, Evaluation No. 18," JPL Publication 15-10, Jet Propulsion Laboratory, Pasadena: https://jpldataeval.jpl.nasa.gov, access: 23 July 2023, 2016.

IUPAC: Task Group on Atmospheric Chemical Kinetic Data Evaluation, edited by: Ammann, M., Cox, R.A., Crowley, J.N., Herrmann, H., Jenkin, M.E., McNeill, V.F., Mellouki, A., Rossi, M. J., Troe, J. and Wallington, T. J.: https://iupac.aeris-data.fr/en/home-english/, access: 29 January, 2024.

Liebmann, J. M., Schuster, G., Schuladen, J. B., Sobanski, N., Lelieveld, J., and Crowley, J. N.: Measurement of ambient $NO_3$ reactivity: Design, characterization and first deployment of a new instrument, Atmos. Meas. Tech., 10, 1241-1258, doi:10.5194/amt-2016-381, 2017.

---

## Author Comment (AC2)

**Reply to RC1**

*In the following, the referee's comments are reproduced (black) along with our replies (blue) and changes made to the text (red) in the revised manuscript. Line numbers refer to those in the initial submission.*

**General Comments:**

The work by Patrick Dewald et al. is relevant to the atmospheric science community, well organized and well written. The quantitative details regarding $NO_3$ reactivity and fractional contribution to biogenic volatile organic compound (BVOC) oxidation appear robustly determined, clear and generally consistent with literature. Their results suggest an important conclusion that $NO_3$ can play a significant role during daytime oxidation which is counter to textbook atmospheric chemistry but not necessarily unheard of. However, there are a couple of points of concern.

We thank the reviewer for the positive assessment of our manuscript and for the provision of helpful comments.

First, the authors are reserving a more in-depth analysis of $NO_3$ oxidation from the same measurements and campaign for a future publication. These details seem highly relevant to information presented here and I question why they are better suited for an independent publication. Doing so removes context which would define the significance of the results.

The main objectives of this work are to identify factors that control the temporal and vertical variability in $k^{VOC}$ and to assign the contribution of VOCs, NO and photolysis to $NO_3$ removal. We refrain from including explicit analysis of VOC and $NO_3$ measurements for the following reasons:

1) The scope of this paper is already well-defined and extensive. We believe that inclusion of the analysis of $NO_3$ and VOC measurements not only conceals the key message but also reduces the readability of the resulting (longer) paper.

2) A rigorous assignment of $k^{VOC}$ to specific compounds is not (yet) possible due to lack of speciated monoterpene measurements at ground level.

3) Ideally, all directly related information from the campaign would be incorporated in one paper. However, as the reviewer will be aware, students, post-docs etc. need first author publications. Expanding this manuscript to cover detailed aspects of $NO_3$-reactivity as related to VOCs would not only make it unreadable but would also preclude first-author papers for the owners (French groups) of the VOC-data and the $NO_3$-measurements. As we already note in the paper, a comparison between calculated (steady-state) and measured $NO_3$ (or $N_2O_5$) mixing ratios above and below canopy level are going to be presented in the future publication along with an assignment of $k^{VOC}$ to measured VOCs.

4) The concerns raised by the reviewer can be addressed with the data provided in the manuscript. We extend the discussion as detailed in the answers below.

Second, it is unclear if the cause for significant daytime oxidative contribution from $NO_3$ is due to reduced photolysis or increased VOC concentration. The answer to this point may be answered in the author's future publication but, again, it appears highly relevant here.

We agree that this point is highly relevant. In order to assess the impact of actinic flux reduction below the canopy, $F^{VOC}$(5.5 m) was calculated using above-canopy values of $J_{NO_3}$ and NO. This would result in an average daytime value for $F^{VOC}$ of 33 %. This value is lower than the actual value of ca. 50 % and the reduction by a factor of 0.66 can be assigned to the attenuation of sunlight by trees and by the tower. However, even 33 % would be significantly higher than 20 % observed in a boreal forest in Hyytiälä, (Liebmann et al., 2018). This difference can be reconciled with higher daytime values $k^{VOC}$ measured in Rambouillet. The increased daytime average of $F^{VOC}$ thus stems from both reduced photolysis below the canopy and presumably from the higher abundance of highly reactive BVOCs. We accordingly extend the discussion in our manuscript in L264:

In order to assess the impact of this effect, we calculated $F^{VOC}$ with above-canopy values of $J_{NO_3}$ and NO. In this scenario, daytime $F^{VOC}$ increases to 33 %, i.e. the reduction of photolysis frequencies increases $F^{VOC}$ by a factor of 1.5. Liebmann et al. (2018) reported a daytime average for $F^{VOC}$ of only 20 % in a boreal forest. Despite the fact that both sites are similarly affected by low NO and high monoterpene levels, this value is still significantly lower than 33 %, which can be reconciled with lower daytime values of $k^{VOC}$ in the boreal forest. The comparatively high daytime contribution of VOCs to $NO_3$ consumption below the canopy thus stems from both reduction in $J_{NO_3}$ and higher values of $k^{VOC}$, latter most likely due to higher concentrations of monoterpenes than in the boreal forest.

**Specific Comments in order of line number:**

1) Lines 100–108: Here, the authors describe the inlets however I did not see any discussion of inlet loss rates or the potential effects on reactivity measurements. I am concerned that the 40 m tall measurements (with a 20 m length and ~ 5 s residence time) may have wall loss of VOCs. Such an effect would result in reduced $NO_3$ reactivity as seen in Figure 1. Some mention of this point would strengthen the validity of these measurements.

During the week when switching between tower and ground measurement, the instrument not only sampled through the tubing (ca. 5 s residence time) but also through the glass flask (ca. 40 s residence time) throughout the diel cycle. The residence time in the PFA tubing is thus only of minor importance compared to the time spent in the flask. From the time before 18 July, no change in $k^{VOC}$ levels was observed when switching between flask and bypass at 03:30 and 19:30 UTC, suggesting that no compounds significantly contributing to $k^{VOC}$ are lost in the glass flask. Given the higher reactivity of uncoated borosilicate glass surfaces compared to PFA and the shorter residence time in the tubing, a significant transmission loss in the latter

appears unlikely. We now clarify this in the manuscript by adding in L118-120:

As the presence of $NO_3$ and $N_2O_5$ in ambient air would bias the measurement, **at nighttime** the air was sampled through a 2 L glass flask (**heated to 35°C**, ~ 40 s residence time) to ensure that ambient $N_2O_5$ is converted to $NO_3$. All radicals including $NO_3$, OH, $RO_2$ and $HO_2$ are lost on the glass walls and thus prevented from reaching the flowtube. **From July 18, air was sampled through the glass flask throughout the diel cycle. Note however that no difference in $k^{VOC}$ levels was observed directly after switching between "daytime mode" (no flask) and "nighttime mode" (sampling through flask) during the period before. This implies that no compounds significantly contributing to $k^{VOC}$ are lost in the glass flask.**

2) Section 3.1: the overall discussion here is clear, but I am left with a question of how the temperature inversion effects the production of $NO_3$, which will contextualize the importance of the resulting $k^{VOC}$. As stated by the authors, the temperature inversions are associated with decreased $O_3$ (and presumed decrease in $NO_3$ production) but increased $k^{VOC}$. The effect of these two counteracting variables is absent from the discussion. I believe such a discussion is needed.

Such a discussion is misplaced in section 3.1 since the variability in $k^{VOC}$ (main point of this section) is independent of the $NO_3$ production rate. The interplay only becomes relevant in the discussion of $NO_3$ mixing ratios. Note that "these counteracting variables" both lead to a reduction in $NO_3$. As indicated in the answer to the general comments, an in-depth analysis of $NO_3$ production rates and mixing ratio will be addressed in a future publication. Instead, we emphasize this point even more than before in L330:

This is not only related to the reduced availability of $O_3$ but also to the increase in $k^{VOC}$, both of which were usually accompanied by temperature inversions.

3) Lines 220–222: The authors state that their analysis (Figure 3) suggests temperature is an important factor that influences $NO_3$ reactivity by BVOC emission. However, I do not find the same conclusion because such a relationship is not apparent in Figure 3. It is clear that the BVOC emission is dependent on temperature (by design) but, as the authors state in the discussion, the relationship between emission and $NO_3$ reactivity is clouded by competing oxidants during the daytime. The conclusion from this discussion appears to be simply that inversion has a strong effect (Figure 3b). I suggest the authors offer a better justification for their conclusion, remove this point, or provide additional VOC data to elaborate.

As already indicated in the manuscript, daytime chemistry affects VOC concentrations. Nevertheless, measured BVOC mixing ratios can clearly correlate with air temperature as shown for example with isoprene in Kalogridis et al. (2014). In spite of the reduced chemical selectiveness of OH compared to $NO_3$, this is even reflected in temperature-correlated OH reactivities in environments dominated by BVOC emissions (Pfannerstill et al., 2021). Consequently, a correlation between air temperature and $NO_3$ reactivity appears plausible. We admit that the daytime correlation between T(5 m) and $k^{VOC}$ is hard to see in Figure 3a, which

is why we now added an inset plotting daytime $k^{VOC}$ (10:00 to 14:00 UTC) against air temperature along with a linear regression. The daytime values of $k^{VOC}$ (black points due to a new color map, see below) are clearly correlated with air temperature (Pearson correlation coefficient r = 0.66). We modified the figure and caption and the text in L220-224 accordingly:

[Figure]

L220: Relative monoterpene emission factors are temperature-dependent and described by $\exp(\beta(T-297\,K))$ with $\beta = 0.1\,K^{-1}$ in forested environments (Guenther et al., 1993), resulting in a strong seasonal variation (Hakola et al., 2006; Vermeuel et al., 2023). **As a consequence, correlations between air temperature and VOC mixing ratios and OH reactivity have been reported (Kalogridis et al., 2014; Pfannerstill et al., 2021). Figure 3a shows that, during the day (black data points), with temperatures varying from 297 to 311 K, an increase in $k^{VOC}$(5.5 m) is observed. The inset contains daytime values measured between 10:00 and 14:00 UTC against air temperature and a linear regression suggests a fair correlation between the two (Pearson correlation coefficient r = 0.66). The expected factor of 4 increase in the emission rate over this range as reported in Guenther et al. (1993) is much larger than the observed change in $k^{VOC}$(5.5 m).**

4) Lines 264–265: The point by the authors that Decker et al. (I am the first author of this cited paper) find $NO_3$ is a major oxidant during daytime for wildfire plumes due to their reduced photolysis rates is incorrect. Instead, the cited work concludes that $NO_3$ is a major daytime oxidant in wildfire plumes because smoke plumes are a huge source of reactive VOCs which

outcompetes photolysis. This point is (admittedly hard to find) most clearly argued for a smoke plume (Castle) with a $jNO_3$ of 0.14 $s^{-1}$ sampled mid-day which is comparable to mid-day photolysis rates presented here (Figure 1). In that cited case, photolysis accounted for ~0.6 % of $NO_3$ loss but the large source of VOCs accounts for the remainder of $NO_3$ loss. The overall conclusion of the cited work is also that $NO_3$ oxidation can be significant during daytime (in the case of biomass burning plumes).

Thank you very much for clarifying this, we have corrected this point in L264/265:

This observation is consistent with even higher daytime VOC contributions to $NO_3$ loss of >97 % reported for sunlit wildfire plumes by Decker et al. (2021) who reconciled their result with VOC mixing ratios that were sufficiently high to outcompete photolysis and NO.

This bring up the question of whether the large daytime $F^{VOC}$ here is the result of reduced photolysis or large BVOC concentrations (or both). As written, the authors appear to suggest that large daytime $F^{VOC}$ is due to reduced sunlight although a definitive conclusion is not clear. The authors acknowledge that $NO_3$ production will also be affected by reduced sunlight, but the effect is not discussed.

A major conclusion of this work is that $NO_3$ is a significant daytime oxidant in this sub-urban forest yet a reason is not clear. Such a conclusion is counter-intuitive based on textbook atmospheric chemistry (as mentioned by the authors). As such, understanding the conditions that cause such a result is highly relevant and important to the atmospheric community. I feel strongly that an interpretation of the cause of a large daytime $F^{VOC}$ would provide highly important context for the author's conclusion that $NO_3$ reactivity is significant during the daytime. I strongly suggest the authors provide further analysis (e.g. $NO_3$ production, information on BVOCs and concentrations) so that the importance of this result is better realized.

We have already addressed these points in the answer to the general comments above.

5) Section 3.6: I have a minor point that (to my understanding) equation 4 is a reactivity, not an oxidation rate, because it is first order and does not include the VOC concentration. Perhaps it could also be called the total loss rate as was done for $NO_3$.

We agree and now call it, in analogy to $NO_3$, loss rate coefficient throughout the whole manuscript.

**Technical Details:**

Line 283: "Recall however, that due its poor" typo here.

Correction made to "due to".

Line 296: "Figure 2 (period p)" seems wrong.

Correction made to "Fig. 7".

Figure 3(a): I suggest to change to x-axis label to something more interpretable by the reader such as "Calculated BVOC emission (unitless?)".
Adjustment made, we changed the x-axis label to "$E_{MT}(T) / E_{MT}(297\ K)$".

Figure color scales: I encourage the authors to use color scales that are readable in grey-scale. This improves accessibility to color blind readers. For example, see Crameri et al. 2020 (https://doi.org/10.1038/s41467-020-19160-7).

Adjustment made, Fig. 2, 3 and S1 have been modified accordingly. We further note in the Acknowledgements:

The Scientific colour map "Berlin" (Crameri, 2023) is used in this study to prevent visual distortion of the data and exclusion of readers with colour-vision deficiencies (Crameri et al., 2020).

Figure markers: to the above point I encourage the authors to use opposing marker styles (such as filled and empty circles) so that the reader does not rely on color to differentiate these markers.

All figures have been modified accordingly.

**References**

Crameri, F., Shephard, G. E., and Heron, P. J.: The misuse of colour in science communication, Nat. Commun., 11, 5444, doi:10.1038/s41467-020-19160-7, 2020.

Crameri, F.: Scientific colour maps (8.0.1), Zenodo, doi:10.5281/zenodo.8409685, 2023.

Decker, Z. C. J., Robinson, M. A., Barsanti, K. C., Bourgeois, I., Coggon, M. M., DiGangi, J. P., Diskin, G. S., Flocke, F. M., Franchin, A., Fredrickson, C. D., Gkatzelis, G. I., Hall, S. R., Halliday, H., Holmes, C. D., Huey, L. G., Lee, Y. R., Lindaas, J., Middlebrook, A. M., Montzka, D. D., Moore, R., Neuman, J. A., Nowak, J. B., Palm, B. B., Peischl, J., Piel, F., Rickly, P. S., Rollins, A. W., Ryerson, T. B., Schwantes, R. H., Sekimoto, K., Thornhill, L., Thornton, J. A., Tyndall, G. S., Ullmann, K., Van Rooy, P., Veres, P. R., Warneke, C., Washenfelder, R. A., Weinheimer, A. J., Wiggins, E., Winstead, E., Wisthaler, A., Womack, C., and Brown, S. S.: Nighttime and daytime dark oxidation chemistry in wildfire plumes: an observation and model analysis of FIREX-AQ aircraft data, Atmos. Chem. Phys., 21, 16293-16317, doi:10.5194/acp-21-16293-2021, 2021.

Guenther, A. B., Zimmerman, P. R., Harley, P. C., Monson, R. K., and Fall, R.: Isoprene and Monoterpene Emission Rate Variability - Model Evaluations and Sensitivity Analyses, J. Geophys. Res.-Atmos., 98, 12609-12617, doi:10.1029/93jd00527, 1993.

Hakola, H., Tarvainen, V., Back, J., Ranta, H., Bonn, B., Rinne, J., and Kulmala, M.: Seasonal variation of mono- and sesquiterpene emission rates of Scots pine, Biogeosciences, 3, 93-101, doi:10.5194/bg-3-93-2006, 2006.

Kalogridis, C., Gros, V., Sarda-Esteve, R., Langford, B., Loubet, B., Bonsang, B., Bonnaire, N., Nemitz, E., Genard, A. C., Boissard, C., Fernandez, C., Ormeño, E., Baisnée, D., Reiter, I., and Lathière, J.: Concentrations and fluxes of isoprene and oxygenated VOCs at a French Mediterranean oak forest, Atmos. Chem. Phys., 14, 10085-10102, doi:10.5194/acp-14-10085-2014, 2014.

Liebmann, J., Karu, E., Sobanski, N., Schuladen, J., Ehn, M., Schallhart, S., Quéléver, L., Hellen, H., Hakola, H., Hoffmann, T., Williams, J., Fischer, H., Lelieveld, J., and Crowley, J. N.: Direct measurement of $NO_3$ radical reactivity in a boreal forest, Atmos. Chem. Phys., 18, 3799-3815, doi:10.5194/acp-18-3799-2018, 2018.

Pfannerstill, E. Y., Reijrink, N. G., Edtbauer, A., Ringsdorf, A., Zannoni, N., Araújo, A., Ditas, F., Holanda, B. A., Sá, M. O., Tsokankunku, A., Walter, D., Wolff, S., Lavrič, J. V., Pöhlker, C., Sörgel, M., and Williams, J.: Total OH reactivity over the Amazon rainforest: variability with temperature, wind, rain, altitude, time of day, season, and an overall budget closure, Atmos. Chem. Phys., 21, 6231-6256, doi:10.5194/acp-21-6231-2021, 2021.

Vermeuel, M. P., Novak, G. A., Kilgour, D. B., Claflin, M. S., Lerner, B. M., Trowbridge, A. M., Thom, J., Cleary, P. A., Desai, A. R., and Bertram, T. H.: Observations of biogenic volatile organic compounds over a mixed temperate forest during the summer to autumn transition, Atmos. Chem. Phys., 23, 4123-4148, doi:10.5194/acp-23-4123-2023, 2023.